# 3D Metamaterials Facilitate Human Cardiac MRI at 21.0 Tesla: A Proof-of-Concept Study

**DOI:** 10.3390/s25030620

**Published:** 2025-01-21

**Authors:** Bilguun Nurzed, Nandita Saha, Jason M. Millward, Thoralf Niendorf

**Affiliations:** 1Berlin Ultrahigh Field Facility (B.U.F.F.), Max-Delbrück-Center for Molecular Medicine in the Helmholtz Association (MDC), 13125 Berlin, Germany; bilguun.nurzed@mdc-berlin.de (B.N.); nandita.saha@mdc-berlin.de (N.S.); jason.millward@mdc-berlin.de (J.M.M.); 2Faculty V, Medical Engineering, Technische Universität Berlin, 10587 Berlin, Germany; 3Faculty II, Berliner Hochschule für Technik, 13353 Berlin, Germany; 4Charité—Universitätsmedizin Berlin, Experimental and Clinical Research Center (ECRC), A Joint Cooperation Between the Charité Medical Faculty and the Max-Delbrück Center for Molecular Medicine in the Helmholtz Association, 13125 Berlin, Germany

**Keywords:** electromagnetic waves, radiofrequency (RF) antenna design, transmit RF arrays, metamaterial, ultrahigh field magnetic resonance imaging (UHF-MRI), cardiac MRI, split ring resonator, numerical simulations

## Abstract

The literature reports highlight the transmission field (*B*_1_^+^) uniformity and efficiency constraints of cardiac magnetic resonance imaging (MRI) at ultrahigh magnetic fields (UHF). This simulation study proposes a 3D Metamaterial (MM) to address these challenges. The study proposes a 3D MM consisting of unit cells (UC) with split ring resonator (SRR) layers immersed in dielectric material glycerol. Implementing the proposed MM design aims to reduce the effective thickness and weight of the dielectric material while shaping *B*_1_*^+^* and improving the penetration depth. The latter is dictated by the chosen array size, where small local UC arrays can focus *B*_1_*^+^* and larger UC arrays can increase the field of view, at the cost of a lower penetration depth. Designing RF antennas that can effectively transmit at 21.0 T while maintaining patient safety and comfort is challenging. Using Self-Grounded Bow-Tie (SGBT) antennas in conjunction with the proposed MM demonstrated enhanced *B*_1_^+^ efficiency and uniformity across the human heart without signal voids. The study employed dynamic parallel transmission with tailored kT points to homogenize the 3D flip angle over the whole heart. This proof-of-concept study provides the technical foundation for human cardiac MRI at 21.0 T. Such numerical simulations are mandatory precursors for the realization of whole-body human UHF MR instruments.

## 1. Introduction

Magnetic Resonance Imaging (MRI) is one of the most advanced non-invasive diagnostic imaging modalities available. Ultrahigh Field (UHF) MRI using static magnetic field strengths of *B*_0_ ≥ 7.0 T is an innovative technology that has propelled biomedical and diagnostic MRI, ultimately leading to better patient outcomes and enhancing treatment planning [1,2,3]. With 7.0 T human MRI now widely used in clinical research, there is increased interest in developing the technology for even higher static magnetic field strengths, as demonstrated in pioneering reports at 9.4 T [4,5,6], 10.5 T [7,8,9,10] and 11.7 T [11,12,13,14]. The expense and the effort of pursuing higher static magnetic field strengths are justified by the sensitivity gains, where the signal-to-noise ratio (SNR) scales with *B*_0_^1.65–1.95^ [11,15]. This increase in SNR will improve the spatial and temporal resolution of MRI and will be key to novel findings and conclusions that can be adapted to current clinical applications. This progress has generated momentum to drive the MR science community to take ambitious steps towards human MRI at 14.0 T [16]. With the implementation of the first 14.0 T whole-body human MR instrument on the horizon [17,18,19,20], initial reports have demonstrated the feasibility of human MRI of the brain [21] and the torso [22] at 14.0 T (*f* = 600 MHz). The UHF-MRI community is already looking further beyond, toward human MR at *B*_0_ > 20 T (*f* = 850 MHz) [23,24]. Early reports include explorations into electrodynamics and radiofrequency (RF) antenna concepts tailored for MRI of the human brain at *B*_0_ = 21.0 T (*f* = 900 MHz) [25,26].

Body and cardiac MRI at 7.0 T has already been included in early application in clinical studies. Yet, more detailed imaging and spectroscopy of the heart and large vessels remain an important goal to improve our understanding of cardiovascular disease [27,28,29,30,31]. The sensitivity gain at 21.0 T facilitates spatiotemporal resolution at a microscopic to mesoscopic level to characterize myocardial tissue and metabolic processes of the heart [29,32]. Key challenges en route to cardiac MRI (CMR) at 21.0 T include the RF wavelength shortening in tissue, efficiency losses of the transmission field (*B*_1_^+^) used for spin excitation, and increased RF power deposition in tissue that may result in an increased specific absorption rate (SAR). At 21.0 T, the uniformity and efficiency of the *B*_1_^+^ used for spin excitation in MRI are challenging due to the short RF wavelength (*λ*) in tissue (*λ* ~ 4 cm), leading to destructive and constructive RF interferences.

Recognizing these challenges, it is conceptually appealing to pursue advanced RF wave propagation for CMR at 21.0 T. One technical solution that could facilitate this is the use of dielectric materials with high permittivity [33]. These materials can be incorporated into the RF antenna design (dielectric resonator) or be used as coupling pads to enhance the *B*_1_*^+^* homogeneity and efficiency [34,35], especially for deep-lying organs such as the human heart. The electric field of the RF antenna causes a displacement current in the dielectric material. This produces a secondary magnetic field, which can aid the primary electromagnetic field (EMF) to enhance the *B*_1_^+^ homogeneity in MRI [36,37,38]. Dielectric materials exhibit electromagnetic resonances because the material acts as both capacitors and inductors, in the sense of a regular circuit where the resonant frequency is determined by the dimensions and the wavelength of the material [39]. The resonant wavelength is in the same order as the thickness of the dielectric pad [39]. One of the earliest applications of high-permittivity materials consists of large bags of water or D_2_O with *ε*_r_ ~ 78 [36,39]. However, the relatively high losses of water and D_2_O [39] render these impractical for use with increasing magnetic field strength, because the resonance frequency and the dielectric permittivity of the material determine the dimensions of the bag. The use of water or D_2_O to fill a bag covering the upper human torso at 900 MHz (21.0 T) would result in a thickness of 38 mm. This size and weight would be impractical in terms of patient comfort and would be a space constraint. Loss-free dielectric fluids provide an alternative solution. Another approach to achieve thin dielectric pads or dielectric resonators includes dielectric materials with very high permittivity (e.g., CaTiO_3_, *ε*_r_ ~ 160, BaTiO_3_, *ε*_r_ > 1000). Reported high-permittivity dielectric materials used to enhance *B*_1_*^+^* homogeneity and efficiency are bulky, impractical for patient comfort, and can degrade over time [36].

Metamaterials (MMs) [33,40] provide a promising alternative for dielectric pads or dielectric resonators. MMs are artificially engineered composite structures enabling properties not found in nature. Single elements of these structures are precisely arranged periodically to achieve the desired electromagnetic properties. These single elements of MMs are referred to as unit cells (UC), and their design dictates the electromagnetic properties. The UC is kept on a subwavelength scale (~*λ*/10) relative to the incident electromagnetic wave to behave as a homogeneous medium. MMs offer a means for controlling the material properties (permittivity and permeability). By engineering the structure and composition of MMs, their electromagnetic properties can be tailored to define the behavior. For instance, MMs can be designed to possess specific permittivity and magnetic permeabilities. The potential of very high dielectric permittivity and permeability materials for the design of MM at UHF is so far untouched. However, if breakthroughs are possible with materials having the described properties, they can be used for specific applications such as magnetic field manipulation or improving MRI. MMs are increasingly being used for the design of novel RF antenna concepts [41,42,43,44,45,46,47,48,49,50,51,52]. MMs have been used to improve the *B*_1_*^+^* homogeneity and efficiency and reduce the SAR for MRI [45,53,54,55,56]. Despite this progress, the feasibility and benefits of 3D MMs have not yet been investigated for MRI so far. The unique capabilities of 3D MMs provide rich opportunities for RF antenna design tailored for cardiac MRI at 21.0 T.

Recognizing this opportunity, the current proof-of-concept study examines the benefits of 3D MMs for CMR at 21.0 T. A 3D UC is designed and tuned to a resonance frequency of 900 MHz (21.0 T). The UC is integrated in an array to form a UC lattice to enhance the RF transmission efficiency for a target region covering the human heart. Depending on the size of the MM array, the *B*_1_^+^ is shaped to increase the RF penetration depth and to minimize the coefficient of variation of *B*_1_^+^ across the human heart. EMF simulations are conducted on a realistic human voxel model to demonstrate the feasibility of human CMR at 21.0 T.

## 2. Materials and Methods

### 2.1. Unit Cell and Metamaterial Design

For the design of the 3D MM, glycerol was used as a dielectric material that exhibits electromagnetic resonances determined by Equation (1):(1)$$f_{dieletric}=\frac{1}{2\pi \sqrt{L_{dielectric}C_{dielectric}}}$$

For a resonance frequency of 900 MHz, the thickness of the glycerol pad needs to be *λ* = 47.11 mm. To achieve a thinner design, we opted for a ~*λ*/4 glycerol pad with a thickness of 15 mm, which results in a resonance frequency of 1.4 GHz. To shift the resonance frequency, square split ring resonators (SRR) [57,58,59] are added to the front face of the glycerol pad to form a 3D MM. The 3D MM design incorporates periodic UC, each measuring 10 × 10 × 15 mm^3^ (Figure 1). Inside the UC, a single layer of SRR consists of two concentric square rings modeled from a perfect electric conductor (PEC) with a width of *w* = 0.3 mm (Figure 1). The diameter of the first and second rings are *r_1_* = 8.3 mm and *r_2_* = 7.3 mm, respectively, with each ring having a *g* = 0.9 mm split gap positioned diametrically opposite. The addition of SRR introduced extra inductance *L_SRR_* and capacitance *C_SRR_* to the system, which resulted in a frequency shift [60]:(2)$$f_{MM}=\frac{1}{2\pi \sqrt{\left(L_{dieletric}+L_{SRR})(C_{dielectric}+C_{SRR}\right)}}$$

The approximate capacitance of the SRR (*C_SRR_)* [61] is determined as a series of the capacitance induced by the split gap (*C_split_*) and the capacitance between the square rings (*C_gap_*) using the parallel plate capacitor Equations of (3) and (4):(3)$$C=\varepsilon ₀\varepsilon ᵣ\frac{A}{d}$$(4)$$C_{SRR}=\frac{C_{gap\;}C_{split\;}}{C_{gap}+C_{split}}$$

The approximate inductance of the SRR [62,63] can be calculated using the inductance of a microstrip square loop wire with Equation (5) below:(5)$$L_{SRR}=\frac{2\mu_{0}\mu_{r}}{\pi }\left[-2r+1.41r-2.41+r\ln\left(\frac{2r}{\frac{w}{2}}\right)\right]-wg$$
with *r* for split ring diameter, *w* for width of traces, and *g* for gap split. For simplicity, mutual inductances between two concentric rings were neglected [64]. By increasing the number of SRR layers inside the UC in the propagation direction (*z*-axis), additional capacitance and inductance are added, which shift the resonance frequency further towards lower values. In this study, the number of layers was increased from 1 to 4 (Figure 1a–e). Maintaining an overall thickness of 15 mm the distance between the SRR layers was 15 mm for 2 layers, 7.5 mm for 3 layers, and 5 mm for 4 layers.

To assess the resonance frequency of the UC, EMF simulations were performed in CST Studio Suite 2020 (CST Studio Suite 2020, Dassault Systèmes, Vélizy-Villacoublay Cedex, France) in the frequency domain solver using UC boundary conditions (*x*, *y*-axis) with two waveguide ports in the z-direction (Z_min_, Z_max_). The reflection coefficient (S_11_-parameter) was assessed at 900 MHz.

To assess the effect of the proposed 3D MMs, a single antenna simulation using a self-grounded bow-tie (SGBT) antenna building block [65] (Figure 1g) was performed on a homogeneous cubic phantom (size = (200 × 200 × 200) mm^3^, *ε*_r_ = 37.5, *σ* = 0.7 S/m). The UC possessing a resonance frequency at 900 MHz was benchmarked against a (i) water pad (*ε*_r_ = 78.4) and (ii) glycerol pad (*ε*_r_ = 50). Therefore, an array of 2 × 4, 3 × 5, 4 × 6, and 8 × 12 of UC was immersed in the glycerol pad (Figure 2 and Figure 3). The maximum SAR_10g_ and *B*_1_^+^ along a center line of the phantom at a depth of 100 mm were assessed. To mimic a realistic scenario, an array of 3 × 5 UC and 4 × 6 UC was implemented on the human voxel model Duke [53] in conjunction with one SGBT antenna placed on the anterior chest wall, and one SGBT placed on the posterior chest wall (Figure 4). The center of the 3D MM and the SGBT was aligned with the center of the human heart. *B*_1_^+^ and maximum SAR_10g_ were assessed for an equal phase and amplitude excitation for a region of interest covering the whole heart.

### 2.2. Cardiac RF Arrays

To constitute cardiac RF arrays, the SGBT building block [65] was adapted to the ^1^H Larmor frequency at 21.0 T (*f* = 900 MHz). The use of water (*ε*_r_ = 78.4) inside the building block resulted in a size of 8.1 × 16.0 × 29.8 mm^3^ at 21.0 T. To better couple the EMF between the SGBT antenna and tissue, a water pad was used. The downside of using water is the losses within the antenna and the pad [39]. To address this shortcoming in an advanced configuration, a loss-free alternative glycerol (*ε*_r_ = 50) was incorporated into the housing of the SGBT building block and the coupling pad. The reduced permittivity of glycerol resulted in an increased building block size of about 20% along each direction (9.5 × 19.2 × 35.7) mm^3^ at 21.0 T. To shape the EMF, a 3 × 5 and 4 × 6 array of UC with three layers of SRR was included below each SGBT building block. The thickness of the water, glycerol, and the MM pad was kept at >15 mm due to the conformity of the human upper torso. As a baseline, a 32-channel RF array using a (i) water pad, (ii) glycerol pad, (iii) 3 × 5 MM, and (iv) 4 × 6 MM array with three layers of SRR was implemented in the human voxel model Duke (Figure 5). A 5-6-5 matrix arrangement was used for the anterior and posterior sections of the baseline RF arrays tailored for covering the upper torso (Figure 5). The distance between the antenna building blocks was set to 45 mm in the left–right direction and 90 mm in the head–feet direction (Figure 5). The positioning of the SGBT antenna, both in the array and with respect to the upper torso, was kept identical for conditions (i)–(iv) above. The size of the building blocks enabled a high-density RF array comprised of 80 elements (Figure 5). For this purpose, a 5 × 8 array of SGBT building blocks was placed on the anterior and posterior chest wall with 35 mm left–right and 55 mm head–feet center distance (Figure 5). The surface area of the upper torso (~250 mm)^2^ enabled the use of the 3 × 5 MM with three layers of SRR for the high-density RF array, using a decoupling distance of 5 mm between the 3D MM (Figure 5g). The positioning of the high-density array was kept identical for the (i) water pad, (ii) glycerol pad, and (iii) 3 × 5 MM setup.

### 2.3. Electromagnetic Field Simulations

Numerical EMF simulations of the RF arrays were performed using the finite integration technique of CST Studio Suite 2020 (Dassault Systèmes, Vélizy-Villacoublay Cedex, France). Broadband excitation (bandwidth: Δ*f*_TX_ = ±50.0 MHz) was applied for a center frequency of *f*_TX_ = 900 MHz. The male human voxel model Duke (body mass index = 23.1 kg/m^2^) [66] was used due to the larger size of this model compared to the female model Ella, which makes CMR more challenging. The model was truncated at the neck and the hips and placed at the isocenter of an RF shield model of a 21.0 T MRI bore. For the EMF simulations, the electrical material parameters of the antennas and the tissue parameters provided by the IT‘IS Foundation [67] were changed to 900 MHz conditions. Channel-wise tuning and matching of the SGBT was performed in Matlab 2019b (Mathworks, Natick, MA, USA) with a lossy series capacitor and parallel inductor. The losses of the lumped elements were considered through the *Q*-factor and equivalent series resistance based on datasheets (atc100c, American Technical Ceramics, Fountain Inn, NY, USA). The results of the EMF simulations and the material and tissue properties were used for the post-processing (Matlab 2019b) to calculate *B*_1_^+^ and maximum SAR_10g_ distributions at an isotropic resolution of 4.0 × 4.0 × 4.0 mm^3^. The number of mesh cells was around 30 million for all RF arrays for setting 20 mesh cells per wavelength. For each RF array, the power absorption matrix considering the losses in the antenna material, radiation losses, losses in the lumped elements of the tuning and matching network, mutual coupling between the antenna, and the power transfer to the human voxel model was assessed.

### 2.4. Optimal Transmit Efficiency

To benchmark the RF array performance independent of any optimization, and to avoid local minima/maxima, the optimal transmit efficiency (TXE) was evaluated for each voxel individually to yield a theoretical electromagnetic performance limit, considering RF coil and coupling losses [68]. The optimal TXE is defined by the maximum (largest eigenvalue) ratio of the MR signal (*B*_1_^+^) to the dissipated RF power of the sample [68]. The calculated optimal TXE maps were assessed and compared within the region of interest (ROI) covering the entire human heart.

### 2.5. Transmission Field Shaping

Static parallel transmission (pTx) is commonly used for transmission field shaping at 7.0 T but becomes less efficient with increasing field strength [22]. Therefore, dynamic pTx was applied using tailored kT points [69]. A series of RF sub-pulses and gradient blips was performed to homogenize the 3D flip angle (FA) (i.e. minimizing the FA coefficient of variation) over the whole heart [70,71,72]. The pulse design problem was solved in Matlab 2019b using the small-tip-angle approximation for a nominal FA distribution of 10° across the whole heart, with an interleaved greedy + local method [70,71,72]. The computation of the solution included a global RF power regularization but no local SAR constraints. A total of 8 kT point pTx pulses were optimized with rectangular-shaped RF sub-pulses, and a total pulse duration of *τ*_total_ = 1.92 ms (8 × *τ*_sub-pulse_ = 100 µs, 8 × *τ*_blips_ 140 µs). An array [200 RF phase settings × #channels] of randomized channel-wise phases as starting points for 200 optimizations was generated and used for the baseline and high-density RF array setups. The obtained FA maps (*FA = γ B*_1_^+^
*τ*) were scaled into *B*_1_*^+^* efficiency maps for an incident power (*P*_In_) of 1 kW. The maximum SAR_10g_ (*P*_In_ = 1 W) of the kT points was evaluated from the sum of the SAR_10g_ distribution for each sub-pulse. The *B*_1_^+^ efficiency map and the maximum SAR_10g_ of an excitation vector with the highest min(*B*_1_^+^_ROI_) from the 200 optimizations were obtained.

## 3. Results

### 3.1. Unit Cell and Metamaterial Design

Using a glycerol pad with a thickness of 15 mm resulted in a resonance frequency of 1.41 GHz (Figure 1a). The incorporation of one layer of SRR led to a frequency shift to 1.05 GHz with S_11_ < −9 dB (Figure 1b). Advancing the design, a second layer of SRR was incorporated into the back face of the glycerol pad (Figure 1c). This further shifted the resonance frequency down to 900 MHz and provided better matching (S_11_ < −19 dB). Adding another SRR layer between the front and back SRR maintained the 900 MHz resonance frequency with S_11_ < −28 dB (Figure 1d). Four layers of SRR resulted in a further frequency shift to 800 MHz with S_11_ < −45 dB (Figure 1e).

The EMF on the uniform phantom revealed lower SAR for the two SRR layers design compared to the three layers, whereas the three layer design showed better *B*_1_^+^ efficiency. The effect of *B*_1_^+^ shaping facilitated by the 3D MM is dependent on the size of the MM array. An MM array larger than the FOV of the antenna resulted in an increased FOV where the power is distributed over the size of the MM array. Our results showed that small local MM arrays can focus the EMF and increase the penetration depth. The phantom simulations showed at 100 mm depth a *B*_1_^+^_100mm_ = 2.2 µT/√kW and max. SAR_10g_ = 9.9 W/kg for the large water pad. The use of a large glycerol pad (Figure 2 and Figure 3) revealed *B*_1_^+^_100mm_ = 2.7 µT/√kW and a lower max. SAR_10g_ = 8.9 W/kg, while a small local glycerol pad with 40 × 60 mm^2^ dimensions yielded *B*_1_^+^_100mm_ = 4.2 µT/√kW and max. SAR_10g_ = 14.8 W/kg. Using small arrays of UC with three layers of SRR inside the large glycerol pad revealed comparable *B*_1_*^+^* efficiency as the small local glycerol pad but with lower SAR (Figure 3). The 4 × 6 MM with three layers of SRR showed the highest *B*_1_^+^ efficiency with *B*_1_^+^_100mm_ = 4.5 µT/√kW and max. SAR_10g_ = 6.2 W/kg (Figure 3). The corresponding *B*_1_*^+^*_100mm_ and max. SAR_10g_ for the other 3D MM setups with three layers is provided in Figure 3. The best SAR efficiency (*B*_1_^+^/√SAR) was achieved with 4 × 6 MM using two layers of SRR (Figure 2) with a *B*_1_^+^_100mm_ = 3.8 µT/√kW and max. SAR_10g_ = 4.2 W/kg. The corresponding *B*_1_^+^_100mm_ and max. SAR_10g_ for the other 3D MM setups with two layers is provided in Figure 2. For a single SGBT antenna, the 4 × 6 MM with three layers provided +69% *B*_1_^+^_100mm_ enhancement and −30% SAR_10g_ reduction compared to the glycerol setup and +18% *B*_1_^+^_100mm_ enhancement and +48% SAR_10g_ increase compared to the 4 × 6 MM setup with two layers.

Next, the 3D MM was placed on the upper torso of the human voxel model Duke (Figure 4) together with an SGBT antenna building block placed on the anterior and posterior chest wall and aligned with the center of the heart (Figure 4). Using a glycerol pad showed a mean (min.) *B*_1_^+^ of 3.9 (0.0) µT/√kW and a maximum SAR_10g_ of 6.3 W/kg. The implementation of the 3 × 5 UCs with two layers revealed 4.4 (0.0) µT/√kW and the 4 × 6 UCs with two layers 4.5 (0.0) µT/√kW, with a maximum SAR_10g_ of 8.2 W/kg and 6.2 W/kg respectively (Figure 4). Using three layers of SRR inside the UC revealed the 3 × 5 MM 4.5 (0.0) µT/√kW and the 4 × 6 MM 4.7 (0.0) µT/√kW (Figure 4). The increased *B*_1_*^+^* efficiency of the three layer design came with an increased max. SAR_10g_ of 11 W/kg (3 × 5 MM) and 7.7 W/kg (4 × 6 MM). Since the improved *B*_1_*^+^* efficiency of the three layer design is desired for CMR at higher *B*_0_, only the three layer design was used for the following cardiac RF arrays implemented on the human voxel model.

### 3.2. Cardiac RF Arrays

The cardiac RF arrays were tuned and matched to S_11_ < −30 dB. For the baseline and the high-density RF array configurations, using a water pad showed ~50% material losses, and ~50% of the power was transferred to the body. The use of glycerol increased the power transfer to the body up to 96% with 0% material losses. The increased size of the glycerol SGBT building blocks and the permittivity change provided improved coupling compared to water. The proposed 3D MM yielded losses and power transfer to the body, which were comparable with the glycerol setups. Using the 3D MM, coupling was further increased due to increased area coverage below the antenna compared to only glycerol. Increasing the number of channels to 80 increased coupling losses due to the smaller distance between the elements of MM. A summary of the losses is provided in Table 1.

To benchmark the RF array configurations on Duke, the |*B*_1_^+^| superposition of the setups was assessed for an ROI covering the entire human heart, where min. *B*_1_*^+^* indicates signal dropouts within the heart. For the baseline RF arrays, the superposed |*B*_1_^+^| maps (Figure 6) revealed the lowest mean(min) *B*_1_^+^ of 0.28 (0.06) µT/√W for the water pad, followed by the glycerol pad setup (mean(min) *B*_1_^+^ = 0.41 (0.09) µT/√W). The glycerol pad showed better performance with min. *B*_1_^+^, which is increased by up to 50%. The 3 × 5 MM with three layers of SRR showed the highest mean *B*_1_^+^ = 0.43 (0.09) µT/√W. The implementation of the 3 × 5 MM did not improve the min. *B*_1_^+^, while the mean *B*_1_^+^ was slightly improved compared to the glycerol setup. The 4 × 6 MM with three layers of SRR showed no *B*_1_^+^ improvement due to higher coupling effects. If the MM is too close to the next MM, the effect of focusing is compromised, and both separate MMs perform as parts of a larger MM, which distributes the EMF in shallow regions under the array. Therefore, for the high-density RF array configuration, only the 3 × 5 MM with three layers of SRR was implemented to manage coupling constraints (Figure 6). The water pad setup provided the lowest mean(min) *B*_1_^+^ of 0.37 (0.07) µT/√W. For the glycerol pad, *B*_1_^+^ = 0.57 (0.12) µT/√W was obtained while the 3 × 5 MM yielded 0.60 (0.13) µT/√W. The higher number of elements in conjunction with the 3 × 5 MM with three layers of SRR resulted in greater min. *B*_1_^+^ compared to the high-density glycerol and water setup (Figure 6).

### 3.3. Transmission Field Shaping

Notwithstanding the enhanced penetration depth and *B*_1_^+^ efficiency facilitated by the high-density RF array, destructive interferences can still occur without proper control of the EMF. To address this constraint, we performed *B*_1_^+^ shaping using dynamic pTx to achieve a uniform excitation of the whole heart without signal dropouts [22,70]. Replacing water with glycerol in conjunction with dynamic pTx improved the transmission field uniformity and efficiency of the 32-channel baseline configurations (Figure 7). The enhanced penetration depth of the 4 × 6 MM with three layers of SRR improved *B*_1_^+^ homogeneity and min. *B*_1_^+^ in the heart. For 200 optimizations, the 4 × 6 MM with three layers of SRR reduced mean CoV by −36% compared to the water setup, by −22% compared to the glycerol setup, and −16% compared to the 3 × 5 MM with three layers of SRR setup. Obtaining an excitation vector with the highest min. *B*_1_^+^ showed that the use of glycerol and the MM resulted in a SAR_10g_ which exceeded the safety limits of 10 W/kg for local body RF arrays [73]. The 3 × 5 MM with three layers of SRR showed the highest SAR_10g_ of 15.2 W/kg, which is in accordance with the results obtained for the single SGBT building blocks. For the 32-channel baseline RF array, the 4 × 6 MM with three layers of SRR showed the highest min. (*B*_1_^+^) = 0.9 µT/√kW with CoV = 16.2% and max. SAR_10g_ = 11.8 W/kg (Figure 7).

Increasing the number of Tx elements to 80 increased the degrees of freedom for *B*_1_^+^ shaping using dynamic pTx. Unlike the baseline 32-channel RF array configuration, a maximum SAR_10g_ ≤ 4.5 W/kg was obtained for the high-density RF array configuration (Figure 8). This meets the safety limits of 10 W/kg for local body RF arrays [73]. Further, improvements in SAR and CoV were obtained for the 80-channel configurations versus the 32-channel baseline configurations. The use of the 3 × 5 MM with three layers of SRR did not improve CoV versus the glycerol setup (Figure 8). The 80-channel array in conjunction with the 3 × 5 MM with three layers of SRR enabled the highest min. (*B*_1_^+^) = 0.9 µT/√kW with CoV = 14.8% and max. SAR_10g_ = 4.5 W/kg (Figure 8). A closer examination revealed that the 3 × 5 MM 80-channel configuration provided a minimum *B*_1_^+^ similar to the 4 × 6 MM 32-channel configuration. However, the CoV was improved by −9%, and SAR_10g_ was reduced by 62% for the 3 × 5 MM 80-channel configuration versus the 4 × 6 MM 32-channel configuration. Compared to the 3 × 5 MM 32-channel configuration, the minimum *B*_1_^+^ was increased by 80%, the CoV was improved by −3%, and SAR_10g_ was reduced by 238% (Figure 8).

## 4. Discussion

This proof-of-concept study demonstrates that 3D MM facilitates human CMR at 21.0 T. Constraints governed by the short wavelength in tissue at 21.0 T was addressed by transmission field shaping facilitated by a novel dielectric 3D MM, which uses arrays of UC comprising three layers of identical square SRR submerged in glycerol [74]. This innovation showed superior performance compared to the conventional dielectric pads that so far have been used to improve EMF coupling and enhance the penetration depth for CMR [75]. Conventional dielectric pads are bulky and not convenient for the patient. Local small dielectric pads are an alternative, but these increase the penetration depth at the cost of increased SAR_10g_. Alternatively, large dielectric pads can be used for EMF distribution to avoid hot spots of RF power deposition, but this approach hampers the penetration depth. Also, dielectric pads may not perfectly conform to the object under investigation. This shortcoming may cause unpredictable air pockets between the pad and the object, which may induce SAR hotspots.

Our findings demonstrate that RF loss mechanisms play a detrimental role at 21.0 T, and need to be carefully considered for the design of RF antenna arrays. Ideally, the total RF input power should be transferred to the upper torso; therefore, unwanted losses such as coupling, radiation, or material losses should be avoided through RF array design considerations. To reduce the losses of the dielectric coupling pads, water was substituted by glycerol in our study. This approach reduced the material losses obtained for a water pad from 51% to 0% and facilitated a power transfer of 96% into the human heart. The proposed 3D MM facilitated a similar power transfer.

To control the EMF, the dimensions of the UC used in our study exceeded the usual rule of <*λ*/10. This rule of <*λ*/10 applies if the MM should be seen as a uniform medium by the incident wave. With this, the MM can possess properties that do not exist in nature (*ε*-permittivity < 0, *µ*-permeability < 0) [76]. Another important factor in MM research is the effective medium ratio (EMR = *λ*/length_UC_), which ideally should be >4 [77,78,79]. In our work, the dimensions of the UC were designed to control the electromagnetic properties of the dielectric material, and an EMR of 4.7 was achieved, which is comparable to previous reports (Table 5 in [80]). Implementing the proposed MM design aims to reduce the effective thickness and weight of the dielectric material while shaping *B*_1_^+^ and improving the penetration depth. The latter is dictated by the chosen array size, where small local UC arrays can focus the EMF field and further increase the penetration depth. A larger UC array can increase the FOV but comes at the cost of a lower penetration depth. Our proposed 3D MM concept consists of various numbers of layers of SRR, which reduces the thickness of the dielectric material. With a greater number of SRR layers, additional capacitance and inductance are introduced, which shifts the resonance frequency to lower values. Further research is still needed to determine the optimal number of SRR layers to reduce the size of the dielectric pad even more, or to achieve an even lower resonance frequency. In addition to varying the number of SRR layers, the liquid in the pads can be changed to shift the resonance frequency, although this could lead to unwanted losses. In the current design, the glycerol of the MM can be inexpensively refilled if degraded, which is an advantage over more expensive dielectric materials with very high permittivity. Additional practical issues are given by the coupling constraints. Placing the local UC arrays closer to each other causes coupling effects, which compromise the effect of RF focusing. To address this, MM can be used as an absorber to avoid coupling [81,82].

Here, we demonstrate the advantages of using 3D MM for MRI, while to date, most reports on MMs generally—and all studies examining MMs for MRI applications—are based on 2D sheets. 3D MMs are commonly used in telecommunication [83,84], optics [85,86], and hyperthermia [87,88]. The latter provides unique opportunities for enhancing the capacity of Thermal Magnetic Resonance (ThermalMR) [25,26,89]. This approach integrates diagnostic MRI with therapeutic thermal intervention in a single RF applicator, and benefits from enhanced RF focusing at higher frequencies.

The 3D MM proposed and evaluated in the current work enables *B*_1_^+^ shaping with increased penetration depth and facilitates human cardiac MRI at 21.0 T. Our approach to controlling the EMF with 3D MM is conceptually appealing for the pursuit of pTx. Currently, pTx is commonly achieved using multi-channel transmit arrays that support exquisite control over the electromagnetic fields by modulating the amplitude and phase used for the excitation of each transmit channel independently. Serious efforts have been devoted to the development of multi-transmit surface RF coil arrays tailored for UHF-CMR [29]. However, pTx systems with 80 independent RF channels exceed the specifications of current commercially available configurations and would be complex and very costly. 3D MMs provide rich opportunities to reduce the complexity and costs of conventional pTx hardware. Instead of modulating the phase and amplitude through the input signal of independent RF channels, 3D MMs can be used for phase control [90,91] while the amplitude can be managed with damping elements. This approach offers the potential to substitute a costly multi-channel RF array system used for spin excitation with a lower number of RF channels, or even with a single transmit RF coil, while EMF management is achieved through 3D MMs.

Physiological motion may constitute a challenge for uniform *B*_1_^+^ excitation of the heart. Cardiac motion commonly deals with the synchronization of the data acquisition with cardiac activity using ECG and other approaches, as documented by a large body of literature [92]. Displacement of the upper torso and the heart due to respiratory motion, cardiac activity, and blood flow may affect the *B*_1_^+^ excitation profile. Since the RF arrays investigated in this study are placed anterior and posterior to the chest, the relative position of the arrays with respect to the upper torso does not change in the presence of bulk motion and is nearly unaltered in the case of respiratory motion. A recent 3-year and inter-day study demonstrated the reproducibility of tailored kT excitation pulses for human cardiac and body imaging [93]. This study reported a remarkable absence of respiratory motion artifacts and demonstrated the feasibility of achieving uniform whole-heart excitation. These results indicate that physiological motion may have a very minor impact on dynamic parallel RF transmission in conjunction with the 3D MMs proposed in our work.

So far, most 2D/3D MMs are passive elements that require transmit elements for excitation. Potential future advancements may involve implementing active MMs, rendering Tx elements redundant, which could reduce the complexity of the RF system even further, improving patient comfort. The potential of 3D MMs is not limited to CMR at 21.0 T, but offers the capacity for advancing MRI in the short wavelength regime, including imaging of the brain, kidney, abdomen, liver, and other target anatomies.

It is a recognized limitation of our study that it does not include real MRI data to further substantiate the feasibility of the proposed 3D MM. At this stage of the development process, we focused on EMF simulations to establish a fundamental understanding of the material’s behavior and to explore its theoretical properties in a controlled manner as a proof-of-concept study. This study aims to provide a metamaterial-based technical solution for the obstacles regarding the B_1_^+^ field inhomogeneity and safety issues governed by the short wavelength regime at 21.0 T. These issues must be resolved as a priority since motion artifacts and other imaging artifacts cannot be effectively considered until these fundamental challenges are addressed. We would like to emphasize that numerical simulations have become a powerful tool for RF technology developments and offer substantial benefits in terms of resources, time, and safety. In this light, our simulations provide a mandatory precursor, technical foundation, and springboard once 21.0 T class human MR instruments become available.

## 5. Conclusions

The pace of discovery of cardiac MRI at 7.0 T is heartening and provides momentum for MRI of the human heart at 21.0 T. Cardiac MRI at 21.0 T faces several challenges, such as the short wavelength regime, pronounced RF losses, and RF power deposition constraints. Going beyond (in Greek ‘Meta’) RF antenna concepts established at lower magnetic field strengths is essential for cardiac MRI at 21.0 T. The use of 3D MMs for MRI at 21.0 T is feasible from an electrodynamic standpoint and provides a technical foundation for explorations into cardiac MRI at UHF. The proposed methods aim to overcome the challenges associated with UHF MRI by improving the efficiency and uniformity of the transmission field while maintaining patient safety. Our work demonstrates the benefits of loss-free 3D MM for RF transmission field shaping at 21.0 T. The 3D MM improved penetration depth and enhanced RF excitation uniformity of the whole heart, free of signal voids, while also meeting the requirements of RF safety.

## Figures and Tables

**Figure 1 sensors-25-00620-f001:**
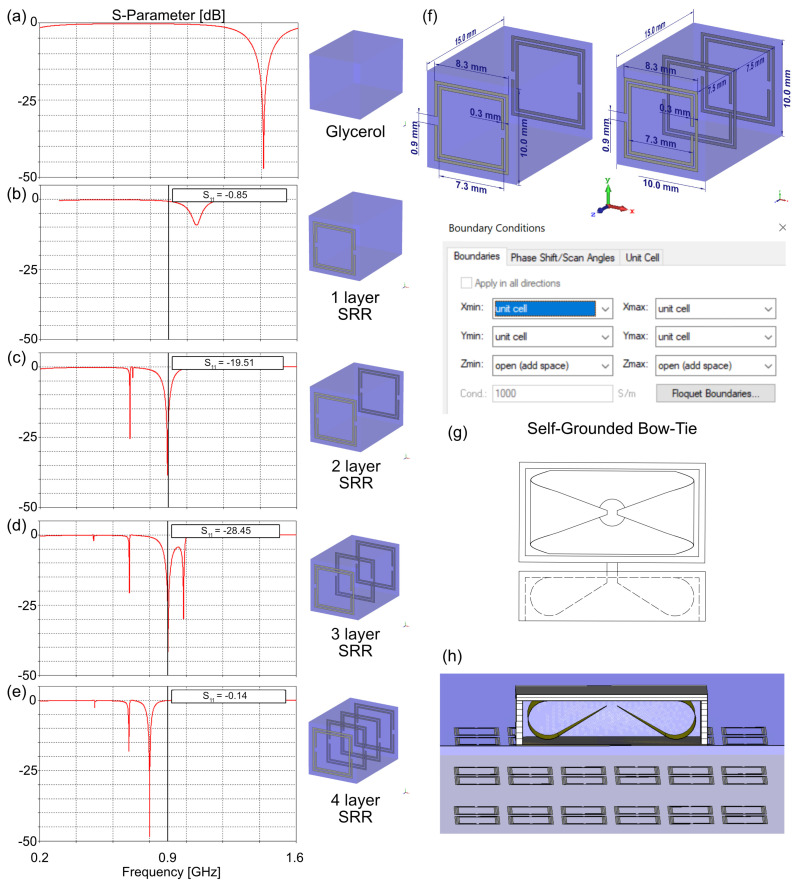
The number of square Split Ring Resonator (SRR) layers was varied from 1–4 using a unit cell (UC) with dimensions 10 × 10 × 15 mm^3^. S_11_–parameter is shown for (**a**) glycerol only with a thickness of 15 mm, (**b**) 1 layer, (**c**) 2 layers, (**d**) 3 layers, and (**e**) 4 layers of SRR immersed in glycerol with a total thickness of 15 mm. (**f**) Design of the 3D Metamaterial with 2 and 3 layers of SRR using UC boundary conditions. (**g**) Schematic of the Self–Grounded Bow–Tie antenna building block with top and side view. (**h**) Positioning of the SGBT antenna shown in (**g**) on top of the 3D Metamaterial containing the UCs shown in (**d**) (center cutting plane).

**Figure 2 sensors-25-00620-f002:**
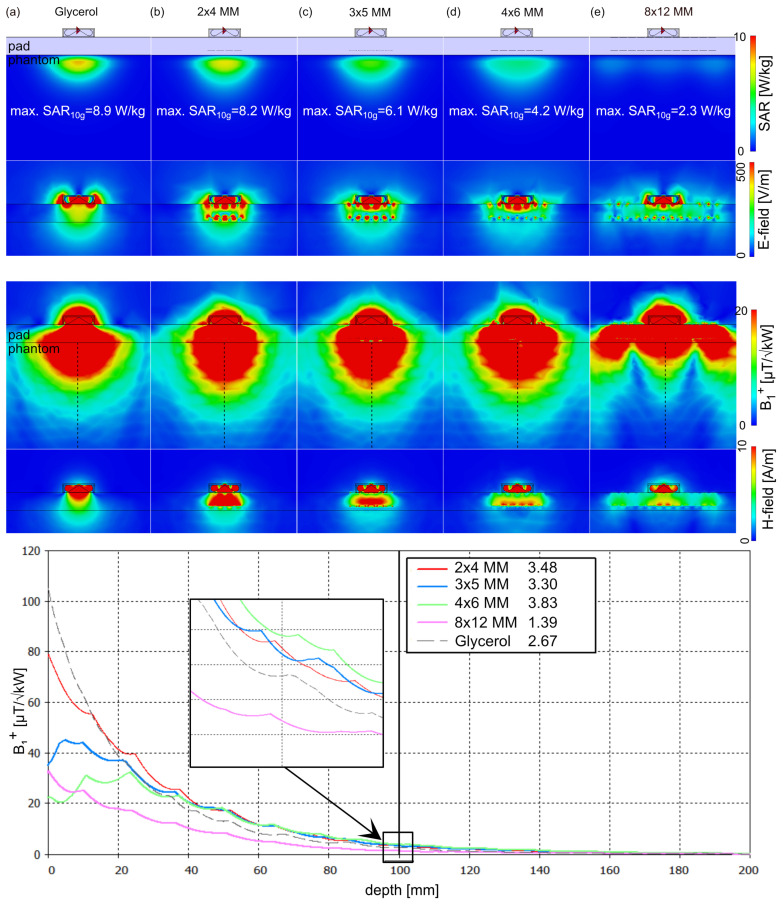
Assessment of the two layer SRR unit cell arrays using EMF simulations of a single SGBT antenna building block placed on a uniform phantom (*ε*_r_ = 37.5, *σ* = 0.7 S/m) using a (**a**) glycerol pad (200 × 200 × 20 mm^3^), and an array of (**b**) 2 × 4, (**c**) 3 × 5, (**d**) 4 × 6, and (**e**) 8 × 12 UC with two layers of SRR placed inside the large glycerol pad. (top) Maps of the Specific Absorption Rate averaged over 10g (SAR_10g_) and E–field. (bottom) Transmission field (*B*_1_^+^) along a center line through the phantom (dashed black line) at 100mm depth (zoomed view) and H–field. As a reference, the baseline configuration using a water pad provided a maximum SAR_10g_ of 9.9 W/kg and *B*_1_*^+^*_100mm_ of 2.2 µT/√kW.

**Figure 3 sensors-25-00620-f003:**
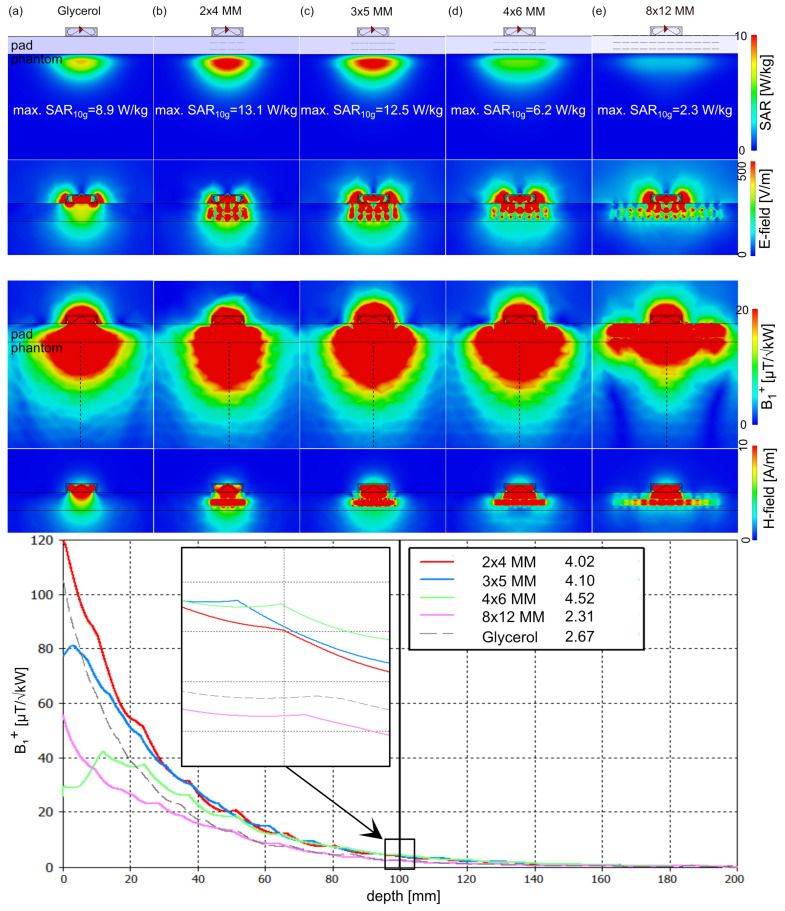
Assessment of the three layer SRR unit cell arrays using EMF simulations of a single SGBT antenna building block placed on a uniform phantom (*ε_r_* = 37.5, *σ* = 0.7 S/m) using a (**a**) glycerol pad (200 × 200 × 20 mm^3^), and an array of (**b**) 2 × 4, (**c**) 3 × 5, (**d**) 4 × 6, and (**e**) 8 × 12 UC with three layers of SRR placed inside the large glycerol pad. (top) Maps of the Specific Absorption Rate averaged over 10 g (SAR_10g_) and E–field. (bottom) Transmission field (*B*_1_^+^) along a center line through the phantom (dashed black line) at 100 mm depth (zoomed view) and H–field. As a reference, the base– line configuration using a water pad provided a maximum SAR_10g_ of 9.9 W/kg and *B*_1_^+^_100mm_ of 2.2 µT/√kW.

**Figure 4 sensors-25-00620-f004:**
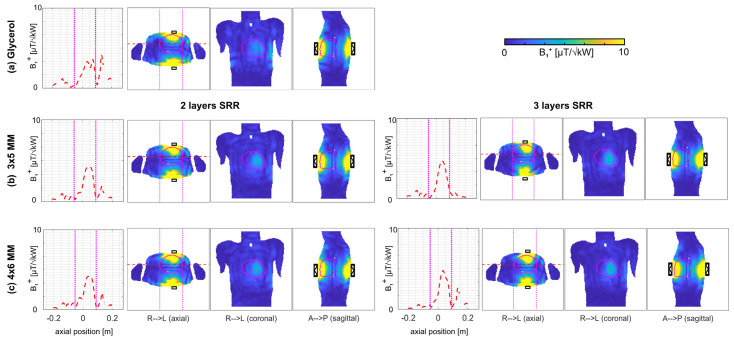
EMF simulations on Duke with one anterior and one posterior SGBT antenna building block (black) aligned with the center of the human heart using (**a**) a glycerol pad, (**b**) a 3 × 5 Metamaterial (MM), and (**c**) 4 × 6 MM inside the glycerol pad deploying two layers (left) of Split Ring Resonators (SRR) and three layers of SRR (right). Transmission field (*B*_1_^+^) maps are shown for the axial, coronal, and sagittal views through the center of the heart (marked in purple). On the left side, the *B*_1_*^+^* along the red dashed line of the axial slice is shown with pink dotted lines depicting the border of the heart.

**Figure 5 sensors-25-00620-f005:**
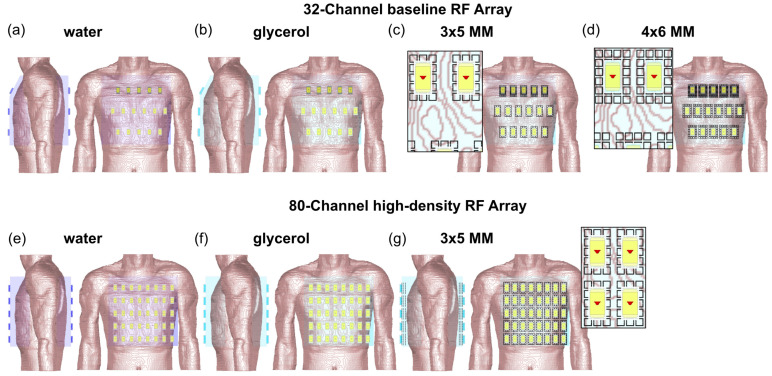
(**a**) Anterior and side view of the design and positioning of the baseline RF array configuration (32 Tx channels, top) and the high–density RF array (80 Tx channels, bottom). (**a**,**e**) A water pad was placed between the SGBT antennas and the chest walls of the human voxel model Duke. The use of glycerol (**b**–**d**,**f**,**g**) instead of lossy water resulted in a 1.2× increase in antenna size compared to (**a**,**e**). The proposed 3D Metamaterial immersed in glycerol was modeled as a (**c**,**g**) 3 × 5 (baseline RF array and high-density RF array) and (**d**) 4 × 6 array (baseline RF array) of unit cells consisting of three layers of SRR. A detailed, zoomed view of the MM setup with the antenna is shown next to the anterior view.

**Figure 6 sensors-25-00620-f006:**
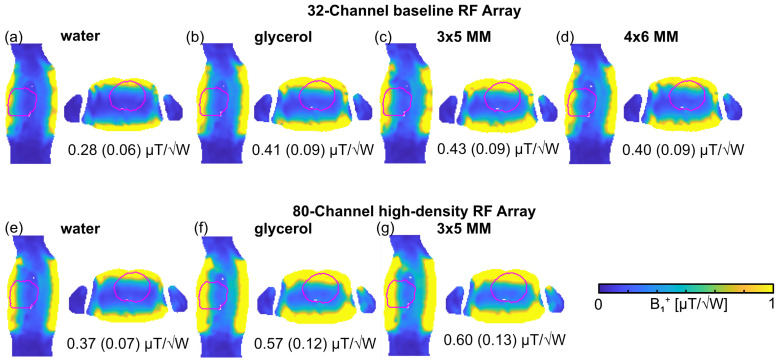
Axial and sagittal views through the center of the human heart (marked in purple) showing |*B*_1_^+^| superposition maps (accounting for sample, coil, and coupling losses) using (**a**,**e**) water pad, (**b**,**f**) glycerol pad, (**c**,**g**) MM with 3 × 5 array of unit cells, and (**d**) MM with 4 × 6 array of unit cells immersed in glycerol. The annotations highlight the mean (min.) *B*_1_^+^ was obtained for an ROI covering the whole heart.

**Figure 7 sensors-25-00620-f007:**
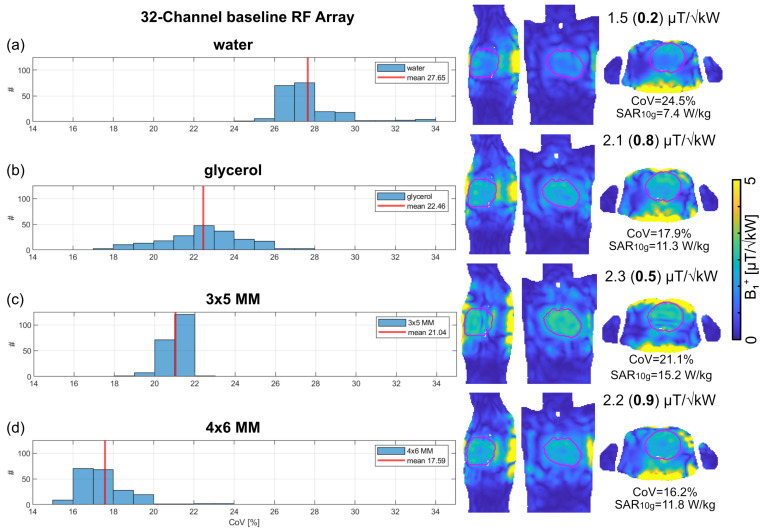
Outcome of transmission field shaping using the 32–channel baseline RF array configuration. Dynamic pTx was performed with tailored kT points to achieve a uniform excitation flip angle (FA) of 10° within the heart. An array of randomized [200 × 32] RF phases was used as starting points for 200 optimizations. The FA maps obtained for the RF pulse design were scaled into *B*_1_^+^ maps. The histogram shows the resulting Coefficient of variation (CoV) of *B*_1_^+^ with the mean values indicated by the red line. *B*_1_^+^ maps for sagittal, coronal, and axial views through the center of the heart (depicted in purple) are shown on the right-hand side for the highest minimum (*B*_1_^+^) for (**a**) water pad, (**b**) glycerol pad, (**c**) 3 × 4 MM, and (**d**) 4 × 6 MM with three layers of SRR immersed in glycerol. The annotations highlight the mean (min.) *B*_1_^+^ efficiency in the whole heart, the CoV of *B*_1_^+^, and the maximum Specific Absorption Rate averaged over 10 g tissue (SAR_10g_).

**Figure 8 sensors-25-00620-f008:**
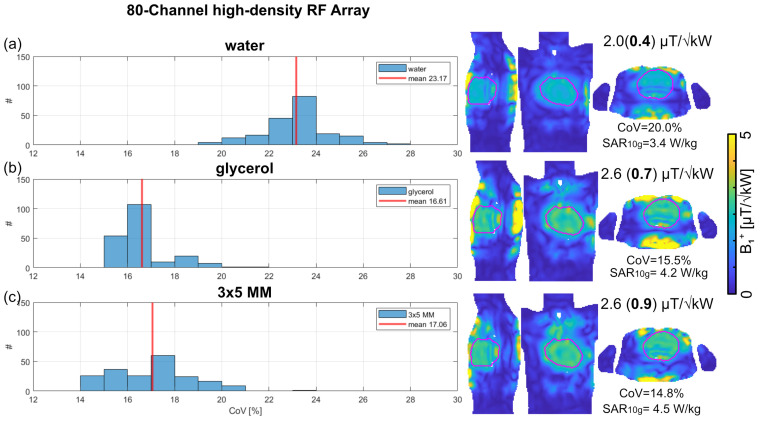
Outcome of transmission field shaping using the 80–channel high–density RF array configuration. Dynamic pTx was performed with tailored kT points to achieve a uniform excitation flip angle (FA) of 10° within the heart. An array of randomized [200 × 80] phases was used as starting points for 200 optimizations. The FA maps obtained for the RF pulse design were scaled into *B*_1_^+^ maps. The histogram shows the resulting Coefficient of variation (CoV) of *B*_1_^+^ with mean values indicated by the red line. *B*_1_^+^ maps obtained for sagittal, coronal, and axial views through the center of the heart (depicted in purple) are shown for the highest minimum (*B*_1_^+^) for (**a**) water pad, (**b**) glycerol pad, and (**c**) 3 × 4 MM with three layers of SRR immersed in glycerol. The annotation highlights the mean (min.) *B*_1_^+^ efficiency in the whole heart, the CoV, and the maximum Specific Absorption Rate averaged over 10 g tissue (SAR_10g_).

**Table 1 sensors-25-00620-t001:** Power absorption matrix accounting for losses in the torso of the human voxel model Duke, mutual coupling between the antenna, losses in the lumped elements of the tuning and matching network (Tu&Ma), radiation losses, losses in the antenna material, and simulation errors (imbalance).

RF Array		Body [%]	Coupling [%]	Tu&Ma [%]	Radiation [%]	Material [%]	Imbalance [%]
32-channel RF array	(i) Water	51.0	0.5	2.0	0.2	44.3	2.0
	(ii) Glycerol	96.1	1.0	2.1	0.6	0.0	0.2
	(iii) 3 × 5 MM	96.1	1.1	2.0	0.5	0.1	0.2
	(iv) 4 × 6 MM	96.1	1.5	1.9	0.4	0.0	0.1
80-channel RF array	(i) Water	44.8	0.7	2.1	0.2	50.8	1.4
	(ii) Glycerol	94.0	3.3	2.0	0.5	0.0	0.2
	(iii) 3 × 5 MM	93.0	5.8	0.6	0.4	0.0	0.2

## Data Availability

The data presented in this study are available on a reasonable request from the corresponding author.

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
