# Peer review of "3D Metamaterials Facilitate Human Cardiac MRI at 21.0 Tesla: A Proof-of-Concept Study"

_sensors, 2025, doi:10.3390/s25030620_

Round 1

Reviewer 1 Report

Comments and Suggestions for Authors

The simulations presented in this article are very impressive but there are some issues that need to be resolved.

1)  The simulations don't include the effect of motion caused by the beating heart.  Please add the effect of motion to the model.

2) A real MRI (non-simulation) image of the new material would be helpful even using a small bore  MRI unit to demonstrate the properties and feasibility  of the new material proposed in this article.

- This simulation study proposes a 3D Metamaterial (MM) to address challenges associated with cardiac MRI at 21.0 Tesla. Ultra high field imaging typically has problems with B1 field transmission and image quality associate with B1 field inhomogeneity.

- This is an important topic because at ultra high field, physical and hardware limitations cause conventional MR imaging pulse sequences to generate artifacts, requiring specialized pulse sequences and new hardware to make use of the gain in signal and contrast.

- Other publications acknowledge the severe problem with B1 inhomogeneity but don’t address how to fix this problem.

- The conclusion that their proof-of-concept study provides the technical foundation for human cardiac MRI at 21.0 T would be consistent if they could provide some real imaging more than just a simulation because simulations don’t address all of the real world problems associated with MRI such as imaging artifacts and motion artifacts.

- The references need to include reference such as this on how to compensate for motion in MRI

https://pubmed.ncbi.nlm.nih.gov/22668237/

- The tables and figures are very thorough showing the results of their simulation but I would like to see additional figure of an MRI acquired at 21.T using their new material.

Author Response

Please see the attachment for the point-to-point response.

Reviewer 2 Report

Comments and Suggestions for Authors

This manuscript presents the challenges and proposed solutions related to cardiac magnetic resonance imaging(MRI)at ultrahigh magnetic fields, specifically at 21.0 Tesla. It includes the model of SSR (Split Ring Resonator) , its equivalent capacitance and inductance with the novelty metamaterial (MM), the simulation electromagnetic field distribution for RF arrays of two or three layers, RF antenna (SGBT), RF loss and so on. Results are very interesting and fundamental achievements could be future used for the higher T (21T) CMR applications.   Here are the key problems and the methods proposed to address them:

Problems:

1.        Transmission Field(B1+)Uniformity and Efficiency Constraints: At ultrahigh magnetic fields, the uniformity and efficiency of the transmission field(B1+) used for spin excitation in MRI are challenging due to the short RF wavelength in tissue, leading to destructive and constructive RF interferences.

2.        RF Power Deposition and Specific Absorption Rate(SAR):  Increased RF power deposition in tissue at higher magnetic fields can result in an increased SAR, which is a safety concern. The authors have made a great progress. We can see it very clearly from figure 2,3 and 4. It is very clearly the authors made a mistake here, the figure captions for figure 2 and 3 is the same. They should be different by the different dielectric permittivity. This could be saw from the explanations from line 286-305.

3.        Line 175, Line 183, ‘glyceral ‘ should be ’glycerol’.

4.         Dielectric Materials Limitations: Traditional high permittivity dielectric materials used to enhance B1+homogeneity and efficiency are bulky, impractical for patient comfort, and can degrade over time. MM is a good choices for SSR. There are still the ways to make a breakthrough by different higher dielectric permittivity material for the electric field building up, of course in principle if we could select also the higher permeability material it will be very useful to build the higher magnetic field especially for higher 21T. so the dielectric material is useful for the electrical field distribution.  The study proposes a 3D MM consisting of unit cells(UC)with split ring resonator(SRR)layers immersed in dielectric material glycerol. This design aims to reduce the effective thickness and weight of the dielectric material while shaping B1+and improving penetration depth.

5.         RF Antenna Design Challenges: Designing RF antennas that can effectively transmit at such high frequencies while maintaining patient safety and comfort is difficult. Using Self-Grounded Bow-Tie(SGBT)antennas in conjunction with the proposed MM demonstrated enhanced B1+efficiency and uniformity across the human heart without signal voids. The study employed dynamic parallel transmission with tailored kT-points to homogenize the 3D flip angle over the whole heart, improving B1+efficiency and uniformity. The optimal TXE was evaluated for each voxel individually to yield a theoretical electromagnetic performance limit, considering RF coil and coupling losses.

6.        The study concludes that the use of 3D MMs for MRI at 21.0 T is feasible and provides a technical foundation for further explorations into cardiac MRI at ultrahigh fields.cThe proposed methods aim to overcome the challenges associated with ultrahigh field MRI by improving the efficiency and uniformity of the transmission field while maintaining patient safety.

Comments on the Quality of English Language

like the comments I did

Author Response

(The authors gave the same response as above.)

Round 2

Reviewer 1 Report

Comments and Suggestions for Authors

I feel that the authors have addressed my previous concerns and made appropriate revisions.